# Vitamin D and Stress Fractures in Sport: Preventive and Therapeutic Measures—A Narrative Review

**DOI:** 10.3390/medicina57030223

**Published:** 2021-03-01

**Authors:** Beat Knechtle, Zbigniew Jastrzębski, Lee Hill, Pantelis T. Nikolaidis

**Affiliations:** 1Medbase St. Gallen Am Vadianplatz, 9000 St. Gallen, Switzerland; 2Institute of Primary Care, University Hospital Zurich, University of Zurich, Pestalozzistrasse 24, 8091 Zürich, Switzerland; 3Department of Health and Life Sciences, Gdansk University of Physical Education and Sport, 80-336 Gdańsk, Poland; zb.jastrzebski@op.pl; 4Division of Gastroenterology & Nutrition, Department of Pediatrics, McMaster University, 1280 Main St W, Hamilton, ON L8S 4L8, Canada; hilll14@mcmaster.ca; 5Exercise Physiology Laboratory, 18450 Nikaia, Greece; pademil@hotmail.com; 6School of Health and Caring Sciences, University of West Attica, 12243 Athens, Greece

**Keywords:** bones, metabolism, sport, sex, stress fracture

## Abstract

There are numerous risk factors for stress fractures that have been identified in literature. Among different risk factors, a prolonged lack of vitamin D (25(OH)D) can lead to stress fractures in athletes since 25(OH)D insufficiency is associated with an increased incidence of a fracture. A 25(OH)D value of <75.8 nmol/L is a risk factor for a stress fracture. 25(OH)D deficiency is, however, only one of several potential risk factors. Well-documented risk factors for a stress fracture include female sex, white ethnicity, older age, taller stature, lower aerobic fitness, prior physical inactivity, greater amounts of current physical training, thinner bones, 25(OH)D deficiency, iron deficiency, menstrual disturbances, and inadequate intake of 25(OH)D and/or calcium. Stress fractures are not uncommon in athletes and affect around 20% of all competitors. Most athletes with a stress fracture are under 25 years of age. Stress fractures can affect every sporty person, from weekend athletes to top athletes. Stress fractures are common in certain sports disciplines such as basketball, baseball, athletics, rowing, soccer, aerobics, and classical ballet. The lower extremity is increasingly affected for stress fractures with the locations of the tibia, metatarsalia and pelvis. Regarding prevention and therapy, 25(OH)D seems to play an important role. Athletes should have an evaluation of 25(OH)D -dependent calcium homeostasis based on laboratory tests of 25-OH-D_3_, calcium, creatinine, and parathyroid hormone. In case of a deficiency of 25(OH)D, normal blood levels of ≥30 ng/mL may be restored by optimizing the athlete’s lifestyle and, if appropriate, an oral substitution of 25(OH)D. Very recent studies suggested that the prevalence of stress fractures decreased when athletes are supplemented daily with 800 IU 25(OH)D and 2000 mg calcium. Recommendations of daily 25(OH)D intake may go up to 2000 IU of 25(OH)D per day.

## 1. Introduction

The term stress fracture is familiar to many doctors, therapists and athletes. Stress fractures, either partial or complete, are common overuse injuries caused by a repetitive submaximal bone loading [1]. The location of the stress fracture varies from sport to sport, but is most commonly observed in the lower extremities [2,3]. They are particularly common in the physically active individuals including but not limited to track and field athletes, long distance runners, dancers, and military recruits [2,3]. The prevalence of stress fractures is estimated to be between 6.5–9.7% among athletes of different sport disciplines [4]. Several studies have noted that an alteration in an athlete’s training program is one of the most significant factors resulting in an injury [5].

It has been theorized that vitamin D (25(OH)D) has a central role in the development of stress fractures. The term 25(OH)D will now be used throughout the manuscript for Vitamin D. Recently, there has been an increased investigation into the relationship of 25(OH)D and stress fractures [6,7]. 25(OH)D deficiency has been shown to be associated with an increased incidence of bone fatigue and stress fractures [8,9,10]. Athletes who experience a stress fracture normally are young and healthy and do not have an underlying metabolic bone disease [11]. Although no evidence exists that athletes have a higher daily requirement of 25(OH)D than the general population, 25(OH)D deficiency in athletes has been linked to a decreased physical performance and a predisposition to stress fractures [12]. Athletes whose bone mineral density is reduced together with a low intake of dietary calcium and low circulating levels of 25(OH)D seem to be at an increased risk of stress fractures [11]. We have a considerable knowledge about the epidemiology of bone overuse injuries and stress fractures in athletes [13,14], bone health of athletes [15,16,17,18], skeletal muscle function in athletes [19] and the influence of 25(OH)D supplementation on bone mass [16], muscle strength [16] and performance in athletes [17,19,20,21]. However, little is known regarding the relationship between stress fractures in athletes and the importance of 25(OH)D [6]. Limited experience suggests that calcium and 25(OH)D supplementation might be helpful [11]. 25(OH)D may protect against overuse injuries such as stress fractures [15,20]. Although several studies have examined the link between 25(OH)D and stress fractures in an athletic population, the aim of this narrative review is to summarize existing findings about 25(OH)D in prevention and therapy regarding stress fractures in athletes and to formulate an up-to-date best evidence synthesis.

Several reviews already reported the importance of 25(OH)D in athletes regarding different aspects such as the influence on performance [20,21], bone health [6], bone health and athletic performance [17], muscle function and performance [19], and the intake of calcium and 25(OH)D in the prevention of stress fractures [15]. Accordingly, a narrative review was conducted according to the definition outlined by Grant and Booth [22]. This method allowed for identification, summation and consolidation of the existing literature, enabling the discovery of potential gaps or omissions in previous bodies of work. In this narrative review, we summarize the current state of knowledge about the connection between 25(OH)D and stress fractures in sport especially for the aspects of prevention and therapy. More specifically, we applied a systematic search and review to the literature search [22] allowing for the best evidence synthesis.

For this purpose, we searched the ‘Scopus’ and ‘PUBMED’ databases for the terms ‘stress fracture-vitamin D-athlete’ from database inception until January 2021. The search of the ‘Scopus’ database lead to 380 articles, the additional search in ‘PUBMED’ lead to 83 articles. Only articles investigating athletic populations were considered for inclusion (Table 1). We included case reports, original papers and review articles investigating athletes. We excluded articles (i.e., studies and case reports) on non-athletes and other types of articles such as notes, conference papers, letters, animal studies, short surveys, editorials, and author responses. For the specific aspects of prevention, we searched ‘PUBMED’ with the terms ‘prevention-stress-fracture-vitamin D’ leading to 90 articles. For the aspect of therapy, we searched with the terms ‘therapy-stress-fracture-vitamin D’ leading to 137 articles in ‘PUBMED’. Ensuring that no relevant publication was missed, we ran web engine searches using Google Scholar and Web of Science databases to screen the first 10 result pages. Any studies that were not part of our initial database search were included.

A total of 180 articles were included in this review. Thirteen themes were identified and synthesized in the sections that follow.

## 2. The Importance of 25(OH)D

25(OH)D is a fat-soluble vitamin and was first discovered in cod liver oil [23]. Since then, 25(OH)D has been identified as an essential vitamin that functions as a precursor steroid for a variety of metabolic [23] and biological processes [23]. 25(OH)D is synthesized either from diet (fortified dairy products or fish oils) or through ultraviolet irradiation of 7-dehydrocholesterol in the skin [23,24]. As soon as it is converted into its biologically active form 1,25-dihydroxyvitamin D [25], it regulates the expression of over 900 gene variants [26]. These gene expressions have a major influence on a large number of health and performance-related aspects such as exercise-induced inflammation, tumor suppressor genes, neurological functions, cardiovascular health, glucose metabolism, bone metabolism and the performance of skeletal muscles [27,28,29,30,31,32,33,34,35,36]. Surprisingly, almost 90% of the world’s population have insufficient 25(OH)D levels [37]. In most of these cases, 25(OH)D deficiency often only causes very unspecific symptoms [38].

## 3. The Importance of 25(OH)D in the Metabolism of Athletes

25(OH)D has many important functions in metabolism [23]. A 25(OH)D deficiency has been linked to increased risk of cardiovascular disease [39,40]. 25(OH)D also appears to be important for certain cardiovascular risk factors [41]. A direct connection between the 25(OH)D level and cholesterol, triglycerides, blood sugar and insulin could be demonstrated [42]. Changes in the lipid profile depending on the 25(OH)D level have already been investigated, but the relationship between lipid metabolism and 25(OH)D has not been fully clarified [39]. In a double-blind and placebo-controlled study with football players during HIIT (high-intensity interval training), however, there was no change in the lipid profile with daily 25(OH)D supplementation of 5000 IU [39]. In another study with professional rowers, 25(OH)D supplementation during exercise showed a significant decrease in cholesterol and a significant increase in 25(OH)D levels [43]. It can be assumed that the improvement in the lipid profile during physical activity depends on the 25(OH)D status [43]. Further, 25(OH)D deficiency seems to be a major problem for the heart in athletes [40,44,45]. In a study by Allison et al. [40] investigating 25(OH)D status over 500 athletes from several different sports. It was found that only 23% of athletes in the study demonstrated sufficient levels of 25(OH)D. 30% were found to have insufficient levels, 37.2% were 25(OH)D deficient and 11% were found to have a severe deficiency. Furthermore, it was found that athletes with a 25(OH)D deficiency had significantly thinner heart walls, smaller ventricular volumes less ventricular mass than athletes without a 25(OH)D deficiency [40].

25(OH)D appears to be important for the ratio of fat to muscle mass in the body [46,47,48,49,50,51]. High 25(OH)D levels also seem to have a beneficial effect on body composition [52]. When 25(OH)D levels are high, body fat is reduced and muscle mass is increased [42,50,52]. Conversely, athletes with high body fat appear to have an increased risk of 25(OH)D deficiency [53]. Likewise, low 25(OH)D levels correlate with a higher body mass index (BMI) [42,47]. However, no connection between bone density and BMI could be demonstrated [49]. In athletes it has been shown that athletes with a high body weight and/or high fat percentage (e.g., ice hockey players) have a higher risk of 25(OH)D insufficiency or a 25(OH)D deficit [51,53]. Conversely, a high 25(OH)D intake, a balanced diet and physical activity outdoor leads to a reduction in the cardiovascular risk situation in boys [54].

## 4. The Importance of 25(OH)D in Bone Metabolism

In addition to the aspects of importance for metabolism previously mentioned, it is undisputed that 25(OH)D is also of central importance for bone metabolism and overall bone health in active people [55,56,57,58]. It is important to know that the bioavailability of 25(OH)D is crucial for bone metabolism, and not the concentration of 25(OH)D in the blood [59]. A large-scale study with over 600 male athletes of various ethnicities showed that the concentration of 25(OH)D in the blood showed no correlation with the bone density at various measuring points, but that the bioavailability of 25(OH)D did [59]. A sufficiently high 25(OH)D intake seems to be important for certain athletes, especially in sports where body weight is a crucial aspect of performance outcomes. In young male jockeys, the daily intake of 800 mg calcium and 400 IU 25(OH)D improved the metabolites of bone metabolism [57]. A large percentage of professional jockeys in particular have a low bone density, a low BMI and a high bone turnover, which is attributed to their low weight and malnutrition [60]. In swimmers, it was shown that daily supplementation with 4000 IU of 25(OH)D was a cheap and efficient measure to increase both the level of 25(OH)D and bone mass [61].

In addition to 25(OH)D, vitamin K is also important for the bones [62]. As with calcium, vitamin K works synergistically with 25(OH)D to regulate bone resorption, bone activation, and bone distribution [63]. Vitamin K carboxylates the newly formed osteocalcin proteins, which are produced in mature bone cells and are strictly regulated by 25(OH)D [64]. Once the protein is carboxylated, it interacts with calcium ions in bone tissue [65] and has a significant impact on bone mineralization, bone formation, preventing bone loss and possibly preventing fractures in women [66,67,68]. However, osteocalcin production is not suppressed if the vitamin K levels are insufficient [69]. This situation facilitates the build-up of non-carboxylated (inactive) osteocalcin proteins in the bone, which leads to a potential increase in calcium release from the bone and the deposition of calcium in soft tissues such as hardening of the arteries [70,71]. Therefore, 25(OH)D toxicity can only occur in the absence of adequate vitamin K levels.

## 5. Consequences of a 25(OH)D Deficiency in Sport—Stress Fractures

25(OH)D deficiency is highly prevalent in both the general [12] and the athletic population [16,71,72], where professional athletes have a high prevalence of 25(OH)D deficiency [73]. The surprisingly high prevalence of inadequate 25(OH)D levels in athletes depends on different factors such as the geographic location, the time of day and year, the local climate conditions, and the sports disciplines (i.e., indoor vs. outdoor) [74]. In the deficient state, an athlete may be at an increased risk for a stress fracture [55].

A prolonged lack of 25(OH)D can lead to stress fractures in athletes [66,67,75,76,77,78,79]. Spontaneous insufficiency fractures are caused by normal or physiological stress on weakened bone [80]. Several studies have shown the relationship between low levels of 25(OH)D and an increased risk of stress or insufficiency fractures [9,81,82]. In a controlled study, it was shown that 25(OH)D insufficiency was associated with 23.3 times the increased risk of fractures of the fifth metatarsal in university soccer players [68]. In a prospective study with 800 Finnish military recruits, a 25(OH)D value of <75.8 nmol/L was identified as a risk factor for a stress fracture [9]. A study investigating 51 males during a 32-week Royal Marines training programme found that recruits with a stress fracture had lower 25(OH)D concentrations [83]. A higher 25(OH)D concentration was associated with reduced stress fracture risk [83]. In 124 persons with a stress fracture, 53 persons had 25(OH)D levels measured within three months of diagnosis and 44 (83.02%) of the 53 persons had a serum 25(OH)D level <40 ng/mL [82]. A prospective study with 1082 Royal Marine recruits during a 32-week training programme showed that 78 recruits (7.2%) suffered a total of 92 stress fractures during this period. Recruits with a baseline serum 25(OH)D concentration <50 nmol/L had a higher incidence of stress fracture than recruits with 25(OH)D concentration above this threshold [7]. Often, the intake of 25(OH)D is insufficient for the training volume of the athlete due to an inadequate recovery, increased bone turnover from repetitive stress and deficiencies in dietary intake. A study investigating 42 male and female cross-country runners showed that 40% of the female and 35% of the male runners reported a history of a stress fracture, and that all of these did not meet the recommended daily energy intake or adequate intakes for calcium or 25(OH)D required for their amount of training [84].

Stress fractures are the result of excessive stress on the bone due to a prolonged and repetitive loading [85,86]. They can also occur as insufficiency fractures due to secondary osteoporosis [87]. Stress fractures are not uncommon in athletes and affect 1% of the general athletic population [88] but can rise up to 20% of all athletes [11]. A study investigating 5201 female Navy recruit volunteers showed that a total of 309 subjects were diagnosed with a stress fracture resulting in an incidence of 5.9% per 8 weeks [81]. A study investigating 118 NCAA Division I athletes showed two stress fractures (1.69%) in a prospective arm and 34 stress fractures (7.51%) in 453 subjects in a retrospective arm of the study [72].

Most athletes with a stress fracture are under 25 years of age [89,90]. Stress fractures can affect every active person, from recreational exercisers to elite athletes [91]. About a third are amateur athletes, two thirds are elite athletes [89]. Long-distance running and sex, in particular being female, seems to be the larger risk factor for sustaining a stress fracture [86,92,93]. The prevalence of stress fractures can rise up to 15% in runners [88].

## 6. Risk Factors or Risk Situations for a Stress Fracture

The majority of stress fractures occur among persons with normal bones, no acute injury and who are undergoing physical activity to which they are unaccustomed [94,95]. Both biological and biomechanical risk factors contribute to the onset of stress fractures [96,97]. Stress fractures have also been shown to frequently occur among persons routinely engaged in vigorous weight-bearing activities such as long-distance running [2,98]. The underlying pathophysiologic process is believed to be related to repetitive mechanical loading of the bone secondary to physical activity, stimulating an incomplete remodelling response [99,100,101].

There are numerous risk factors for stress fractures that have been identified in the literature [102]. 25(OH)D deficiency is, however, only one of these many factors. Well-documented risk factors include female sex, white ethnicity, older age, taller stature, lower aerobic fitness, prior physical inactivity, greater amounts of current physical training, thinner bones, cigarette smoking, 25(OH)D deficiency, iron deficiency, menstrual disturbances, and inadequate intake of 25(OH)D and/or calcium [14,103]. Table 2 presents the potential risk factors for a stress fracture.

## 7. Symptoms of Stress Fractures

An athlete with a stress fracture typically reports localized pain that gradually worsens, most commonly in the lower extremity [94,95,99,104,105]. Additionally, athletes experiencing a stress fracture have also reported pain that is aggravated by physical activity and relieved by rest [14]. Athletes with a stress fracture usually recount a history of a recent increase in physical activity or the beginning of a new activity or some other change in their routine [13]. They complain about a nagging and aching pain that is felt deep within the foot, toe, ankle, shin, hip, or arm. A pain localized in a particular area, such as the foot, ankle, or hip, appears in the evening and is often associated with a stress fracture, even if the pain is not debilitating during sports activities. Pain that resumes or remains constant despite taking time off to rest and/or using rest, ice, compression and elevation may be due to a stress fracture [95,105,106].

## 8. Diagnosis of a Stress Fracture

The diagnosis of a stress fracture is based on history and selected diagnostic imaging [102]. The early detection of a bone stress injury before it becomes stress fracture is essential [97].

The evaluation of a runner with a suspected stress fracture needs to include completing an appropriate history and physical examination [96,97]. Both the history and the accurate physical examination of the athlete provide the foundation for diagnosing a stress fracture [96,97].

The affected area to test should be examined for pain, tenderness, and swelling, as well for as any associated risk factors for a stress fracture such as weak muscles and/or bone misalignments [13,107]. Accurate palpation elicits localized tenderness over a bone. Additionally, localized swelling and erythema may be observed [13].

Simple clinical tests can assist in the diagnosis, but more definitive imaging tests will eventually be needed to be conducted when a stress fracture is suspected [14]. In case of positive signs during a physical examination, radiographs are indicated for confirmation of diagnosis. When a stress fracture is suspected, a plain radiography should be obtained initially and, if negative, may be repeated after two to three weeks for greater accuracy [13]. However, radiologic signs depend on the time from the onset of symptoms and the type of bone affected. Stress fractures often do not show up on X-ray right away. Radiographic findings may include early lucent zones, periosteal new bone formation, focal sclerosis, endosteal callous, or later fractures or cortical cracks [105,106].

At the onset of symptoms, radiographs may be negative, and radiologic signs, if they become evident, may take several weeks to evolve [105,106,108,109]. Continued pain may warrant advanced imaging, such as scintigraphy or magnetic resonance imaging (MRI) [110]. While they have clinical specificity, radiographs lack sensitivity. Bone scans, on the other hand, have a superior sensitivity, but they lack specificity [111,112], and should not be used alone to make the diagnosis of a stress fracture [112]. Therefore, other types of diagnostic testing such as MRI, computerized tomography (CT) scans, ultrasound, or Technetium-99 bone scans may be used to confirm a suspected stress fracture diagnosis [97]. X-ray can be used to detect older stress fractures that have partially healed, and/or stress fractures that have progressed to a non-union (hairline) or a displaced fracture. If an urgent diagnosis is needed, triple-phase bone scintigraphy or MRI should be considered. Both modalities have a similar sensitivity, but MRI has a greater specificity [13].

## 9. Stress Fractures and Type of Sport

Stress fractures seem to be largely dependent on the type of sport being undertaken [79,86,107,113,114,115,116]. Certain sport disciplines seem to lead to stress fractures more often than others. A study analyzed the distribution by type of sport on the basis of 671 stress fractures in 11,778,145 athletes [116]. Stress fractures occur more frequently in women and in sports such as cross-country, gymnastics, athletics and basketball [116]. There are comparable results in other studies. The following distribution was found in a collective of almost 15,000 athletes: basketball (21.3%), baseball (13.7%), athletics (11.4%), rowing (9.5%), football (8.4%), aerobics (5.3%), and classical ballet (4.9%) [89]. A study examined 279 basketball players over the period from 2009 to 2013. 25(OH)D deficiency was found in 73.5% of all players and 118 players (42.3%) suffered at least one fracture during this period [75].

Athletes in endurance sports such as running, cycling and swimming often have a lower bone density than ball and strength athletes [86,117]. The bone density of runners, cyclists and swimmers can even be lower than that of inactive people [117]. The deep bone density can increase the risk of stress fractures, especially because endurance athletes train and compete for years [117].

Long-distance runners are more likely to suffer a stress fracture of the lower extremity [97,107]. Swimmers are more likely to suffer stress fractures of the ribs than athletes in other sports [114]. In addition to swimmers, rib fractures are also more common in rowers [118]. The rowers consider low calcium, low 25(OH)D, eating disorders and low testosterone as risk factors [118].

## 10. Localization of Stress Fractures

The lower extremity seems to be more affected in general for stress fractures [68,79,86,89,90,119,120,121,122]. The localization of stress fractures by body region also seems to depend on the type of sport. A study analyzed the distribution by type of sport on the basis of 671 stress fractures in 11,778,145 athletes [116]. The lower extremity represents ~80–95% of all stress fractures and the increased popularity of endurance running has contributed to the tibia (49% prevalence) replacing the metatarsals (9%) as the most common location for lower extremity stress fractures [123].

The metatarsals and the tibia are the most frequently affected locations [79,116], especially in basketball players [116]. There are comparable results in other studies. In a collective of almost 15,000 athletes the following distribution was found: tibia (44.1%), ribs (14.1%), metatarsals (12.9%), olecranon (8.7%) and pelvis (8.4%) [87]. Stress fractures of the tibia can also occur as anterior mid-tibial cortex stress fractures [124] and should not be mistaken as medial tibial stress syndrome [78].

Apart from the rather frequent locations, stress fractures can also occur in rare locations [125,126,127]. A rare location for a stress fracture on the lower extremity is the pelvis [113,126], the patella [128,129] and the femoral neck [127] and on the upper body the upper arm [130,131]. In rare cases, also stress fractures of the sacrum can occur [132,133]. In certain very rare cases, stress fractures in the lower extremity can occur bilaterally, such as the patella [128,129], the lower leg [121,134], the tibia [135] or the thigh [136].

In the lower extremity, depending on the study and the group of athletes, the tibia seems to be most frequently affected before the metatarsals. Risk factors for a stress fracture of the tibia are an earlier stress fracture, an increase in the intensity or the duration of the training, insufficient technique during training, insufficient equipment during training, the so-called [88,117] ‘female athlete triad’ [137], a deficit in the intake of calcium and 25(OH)D, as well as an insufficient intake of calories [138].

## 11. Sex and Stress Fractures

As stated previously, there seems to be a difference between the sexs regarding the prevalence of stress fractures [114]. Women are more likely to suffer a stress fracture than men [139,140]. It can be assumed that low bone density due to low 25(OH)D intake and low 25(OH)D levels is the main cause [141], likely from inadequate dietary intake to meet normal physiological needs [11,142]. In contrast to men, women are at risk of a female athlete triad, which is more likely to suffer from stress fractures [143]. The ‘female athlete triad’ includes menstrual disorders, disturbed eating behavior and osteopenia/osteoporosis [144,145], leads to premature loss of bone mass with subsequent osteoporosis [146] and stress fractures [147]. It is assumed that a 25(OH)D deficiency and insufficient calcium intake in women are the main reasons for a decrease in bone density [148]. Up to 50% of female athletes have disordered eating making them at risk for functional hypothalamic amenorrhea [149].

The ‘female athlete triad’ is more common among young female athletes [150]. Up to a quarter of young women athletes have menstrual disorders [151]. In women, increased energy consumption and/or reduced energy intake as well as an eating disorder are the cause of a loss of bone mass [140,152,153,154]. The energy deficit leads to a decrease in estrogen and a change in bone metabolism [140]. However, the definition of the female athlete triad has been recently been redefined as relatively energy deficiency in sports (RED-S) [155]. This broadened syndrome is defined as the ‘*impaired physiological functioning caused by relative energy deficiency and includes*, *but is not limited to*, *impairments of metabolic rate*, *menstrual function*, *bone health*, *immunity*, *protein synthesis and cardiovascular health*’ [156] (p. 687). Accordingly, Mountjoy et al. theorized that the main contributing factor of RED-S is low energy availability [156]. Furthermore, RED-S has been expanded to include male athletes and endocrine dysfunction, metabolism, energy availability, psychology and impact on health and performance [157]. However, the core principle remains the same, insufficient availability of 25(OH)D and energy availability lead to increased risk of bone injuries, including stress fractures.

To reduce the risk of stress fractures in women, female athletes should consume enough calories with their diet, take in sufficient micronutrients such as vitamins and minerals, and perform physical training that increases bone density, such as strength and jumping training [158,159,160]. Women who participate in regular physical activity and exercise have been shown to have higher bone density in certain body regions than inactive women [159]. It is interesting to note that men and women experience different fracture patterns with regard to location. Pelvic fractures seem to occur more frequently in women and more frequently in female long-distance runners [113]. In young women, a connection between 25(OH)D intake and pelvic bone density has been demonstrated [161]. Certain sports are likely to be very beneficial for the bone density of the vulnerable region of the pelvis. It has been shown that figure skaters have increased bone density in the area of the pelvis and legs, most likely due to the heaped landing on their feet after the jumps, and an altered force transference owing to the wearing of ice skates [162].

Stress fractures can occur in both younger and older athletes. A case report describes a 52-year-old ultra-runner who suffered a vertebral fracture as part of a vertebral hemangioma [76]. In the further course of the disease, osteopenia-caused by a 25(OH)D deficiency-was found to be the cause of the fracture [76]. If there is a pronounced deficiency in micronutrients such as 25(OH)D in young athletes, growth and development can be severely disturbed [77]. A rare reason for a stress fracture in a runner could also be hypophosphatasia [163]. On the basis of these findings, various professional associations demand that athletes have to take in enough calcium and 25(OH)D during intensive training [164].

## 12. Treatment of Stress Fractures

Most stress fractures readily heal following a period of modified loading (i.e., reduction in training) and a progressive return to running activities. As the recurrence rate of bone stress injuries is high, it is crucial to address their underlying causative factors [107].

The treatment involves first determining if the stress fracture is of a higher or a lower risk [14]. These two risks are distinguished by anatomical location and whether the bone is loaded in tension (high risk) or compression (lower risk) [14]. Overall, treatment of a stress fracture consists of activity modification, including the use of non-weight-bearing crutches if needed for pain relief. Analgesics are appropriate to relieve pain, and pneumatic bracing can be used to facilitate healing. After the pain is resolved, and the examination shows improvement, patients may gradually increase their level of activity. Lower risk stress fractures can be initially treated by reducing loading on the injured bone by reducing activity or substituting other activities [14]. Higher-risk stress fractures should be referred to an orthopedic surgeon [14]. Surgical consultation may be appropriate for patients with stress fractures in high-risk locations, non-union, or recurrent stress fractures [13].

Also prescribing calcium and 25(OH)D supplementation may help with fracture healing in subjects stress fractures who may have an unrecognized hypovitaminosis-D which-if left untreated-may delay fracture healing [165]. In patients in whom 25(OH)D deficiency is a concern, serum 25(OH)D level is the appropriate screening test, with therapeutic goals for bone health being at least 50 nmol/L and may be as high as 90 to 100 nmol/L [166].

## 13. Prevention of Stress Fractures

The use of 25(OH)D in the prevention of stress fractures is one part. Proposed strategies to prevent stress fractures include modifications of the training [14], such as gradual training adjustments, gradual progressive initiation of vigorous physical training [94,104], reducing impact-related forces and increasing the strength and/or endurance of local musculature [107], recovery periods with no running after 2–3 weeks of training [99,128,167], use of proper running shoes [99] use of shock-absorbent insoles [14], use of orthotic shoe inserts [168], use of foot orthoses [169], adherence to an appropriate diet (i.e., including adequate caloric intake, calcium, and 25(OH)D) [14,96], and treatment of abnormal menses in women [104].

To prevent stress fractures, modifiable causes and risk factors (e.g., nutrition) must be identified [96]. Risk factors for exercise and sports-related injuries, including stress fractures, are commonly categorized as intrinsic or extrinsic (Table 2). Intrinsic factors are characteristics of the individual exercise or sports participant, including demographic characteristics, anatomic factors, bone characteristics, physical fitness, and health risk behaviors. Extrinsic risk factors are factors in the environment or external to the individual participant that influence the likelihood of being injured, such as equipment used or type of sport.

It is also recommended to get enough calcium and 25(OH)D [170,171], optimize bone health, adequate nutrition (i.e., adequate calcium and 25(OH)D) together with appropriate weightbearing exercise and strength training are necessary throughout life [18].

Another approach could be the determination of bone mineral density using dual-energy X-ray absorptiometry to assess the risk for a stress fracture [172,173,174]. It has been shown for female athletes that lower bone density was a risk factor for stress fractures in track and field athletes [172]. Furthermore, bone area and cortical thickness at the tibia were identified as altered both in women with menstrual disturbances and in women with stress fractures [174].

## 14. 25(OH)D Supplementation as a Preventive Measure

25(OH)D and calcium are essential for bone health [175]. It is known that athletes with a lower 25(OH)D concentration had a higher risk of a stress fracture [68] and a higher 25(OH)D concentration was associated with a reduced risk for a stress fracture [83]. Therefore, prophylactic 25(OH)D supplementation seems a valid injury risk mitigation strategy [83,118,176].

Very recent studies showed a positive trend in 25(OH)D concentrations from higher doses of supplementary 25(OH)D and a possible benefit for bone health when 25(OH)D intake was combined with calcium intake [170,171]. It is therefore recommended that 25(OH)D should be determined early and regularly in high-risk athletes [177]. Athletes should have an evaluation of the 25(OH)D -dependent calcium homeostasis based on laboratory tests of 25(OH)D, calcium, creatinine, and parathyroid hormone [74]. Athletes with a high risk of a 25(OH)D insufficiency, such as indoor athletes, must be carefully reviewed [178]. 25(OH)D levels above 50 ng/mL are required for athletes to achieve maximal physical performance [74]. 25(OH)D levels below 40 ng/mL are highly associated with an increased prevalence of stress fractures [82]. 25(OH)D levels lower than 30 ng/mL may lead to defects in bone mineralization (e.g., osteomalacia) and impaired muscular function such as reversible myopathy [74]. It has been shown that an insufficient serum 25(OH)D level lower than 30 ng/mL was associated with a statistically significantly increased risk of a stress fracture of the fifth metatarsal [68]. Although there is still a debate about the appropriate levels of 25(OH)D, studies have suggested a minimal level of 32 ng/mL [36].

Preventative measures for spontaneous insufficiency fractures include—among others—the supplementation of 25(OH)D in athletes with 25(OH)D insufficiency [80]. It is recommended to treat athletes with insufficient or deficient 25(OH)D levels with 25(OH)D [16]. In case of a 25(OH)D insufficiency, normal blood levels of ≥30 ng/mL may be restored by optimizing the athlete’s lifestyle and, if appropriate, by oral substitution of 25(OH)D. This concentration of ≥30 ng/mL is associated with a protective effect and an enhancement of physical performance [74]. A very recent pilot study showed a significant decrease in stress fractures from 7.51% to 1.65% with 25(OH)D supplementation [72].

The amount of 25(OH)D intake required is highly variable depending on many factors, including sun exposure, and therefore many recommendations have been made for daily 25(OH)D intake requirements [166]. Due to its effect on the calcium metabolism, the supply of 25(OH)D should have a beneficial effect on damage caused by overuse such as stress fractures [20]. The supplementation of 25(OH)D can reduce both markers of bone formation and resorption and the decline of 25(OH)D [175]. Especially runners who have already had a stress fracture are at high risk of low bone density [179]. For certain fractures, the preventive use of 25(OH)D is considered [180]. Up to now it has not been proven that the increased intake of calcium and 25(OH)D is useful for the prevention of stress fractures [11]. However, it has been shown that the intake of more than 1500 mg calcium per day reduces the frequency of stress fractures in female athletes [15]. In female Navy recruit volunteers, daily intake of 2000 mg calcium and 800 IU 25(OH)D reduced the risk of stress fractures significantly [81]. The calcium and 25(OH)D group had a 20% lower incidence of stress fractures than the control group (5.3% versus 6.6%, respectively). Recommendations of daily 25(OH)D intake may go up to 2000 IU of 25(OH)D per day [166]. A target for prevention of stress fractures would be a serum 25(OH)D concentration of 40 ng/mL or greater, achievable with 4000 IU/d of 25(OH)D supplementation [181].

In addition to stress fractures, a 25(OH)D deficiency also leads to a reduction in physical performance [182] and muscle strength [53]. A study with young football players showed that a lack of 25(OH)D led to reduced muscle strength, reduced mobility and a reduction in the ability to jump and sprint [182].

Calcium seemed also to be of importance regarding prevention of stress fractures. A randomized double-blind, placebo-controlled study on 5201 female Navy recruit volunteers investigated whether a calcium and 25(OH)D intervention could reduce the incidence of stress fractures in female recruits during basic training. It was shown that the calcium and 25(OH)D group after daily supplementation of 800 IU 25(OH)D and 2000 mg calcium had a 20% lower incidence of tibial stress fractures than the control group [81]. Two studies suggested that female athletes and military recruits who consumed daily, more than 1500 mg of calcium showed the largest reduction in stress fractures [15]. A case report of a 56-year-old man with an un-displaced stress fracture in the femoral neck and a 25(OH)D <10 nmol/L showed a complete reversal of ‘osteoporosis’ within 14 months with 25(OH)D and calcium supplementation [122]. It has also been shown that recruits with low serum 25(OH)D take longer to recover from stress fractures [10].

Apart from the supplementation of 25(OH)D, outdoor training seems to be of importance. A study with female runners showed that distance runners maintained sufficient 25(OH)D status when training occurs outdoor in latitude (30.4° degrees north in the study) where 25(OH)D synthesis occurs year-round to reduce the risk for 25(OH)D deficiency [183].

## 15. Limitations of the Present Review

Several limitations were noted during this review. As we undertook a narrative review, we did not critically appraise the available literature as would have been done in a systematic review. Secondly, we included lower level of evidence studies such as case series and case reports, in this review. Thirdly, the current review utilized only two databases to perform our literature search, however, we performed Google Scholar and Web of Science searches to ensure we did not miss any relevant studies. However, the current review does present an up-to-date synthesis of the best evidence, building on previous existing literature.

## 16. Conclusions

Stress fractures affect around 20% of all athletes where most affected athletes are under 25 years of age. Stress fractures can affect every active person, from weekend athletes to top athletes. Stress fractures are common in specific sports disciplines. The lower extremity is more affected than the upper extremity by stress fractures in the tibia, metatarsals and pelvis. Athletes should have an evaluation of 25(OH)D-dependent calcium homeostasis based on laboratory tests of 25(OH)D, calcium, creatinine, and parathyroid hormone. In the case of 25(OH)D insufficiency, normal 25(OH)D levels of ≥30 ng/mL may be restored by optimizing the athlete’s lifestyle and, if appropriate, the oral substitution of 25(OH)D. Very recent studies suggested that the prevalence of stress fractures decreased when athletes are supplemented daily with 800 IU 25(OH)D and 2000 mg calcium. Recommendations of daily 25(OH)D intake may go up to 2000 IU of 25(OH)D per day. Future studies might investigate the relationship between 25(OH)D and bone mineral density using dual-energy X-ray absorptiometry for athletes with stress fractures and/or athletes with an increased risk of a stress fracture.

## Figures and Tables

**Table 1 medicina-57-00223-t001:** Summary of inclusion and exclusion criteria.

	Inclusion Criteria
1	Studies investigating sport, exercise or physical training
2	Investigating the role of 25(OH)D or make reference to the supplementation, ingestion or use of 25(OH)D
3	Studies investigating bone health, stress fractures, with reference to prevention and/or rehabilitation
4	Study designs including RCTs, case-control, cross-sectional and, case series and reports
5	Human participants
	**Exclusion Criteria**
1	Studies in non-athletic or non-physically active populations
2	Animal studies
3	Study designs including letters, editorials, conference proceedings, short surveys and author responses

**Table 2 medicina-57-00223-t002:** Potential risk factors for stress fractures.

Intrinsic Factors	
Demographic characteristics	Female sex
	Age, with athletes over 40 and under 18 most at risk
	Race (other than white)
Anatomic factors	High foot arches
	Uneven leg and/or foot alignment
	Flat feet (pes planus)
	Knock-knees
	High quadriceps angles
	Leg length discrepancies
Bonce characteristics	Geometry
	Low density
	Uneven leg and/or foot alignment
Physical fitness	Lower aerobic fitness
	Lower muscle strength
	Lower muscle endurance
	Lower flexibility
	Body composition
	Body stature
Health risk behaviors	Sedentary lifestyle
	Tobacco use
	History of injury, stress fracture
	Low calcium intake
	Low protein intake
	High caffeine intake
	Prolonged intake of certain medicaments or drugs
**Extrinsic Factors**	
Type of activity/sport	Track and field
	Dance
	Soccer
	Basketball
	Military basic training
Physical training	High amount of training
	High duration of training
	High frequency of training
	High intensity of training
Equipment	Shoes
	Boots
	Insoles
	Orthotic inserts
Environment	Road
	Trail
	Track

## Data Availability

Not applicable.

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
