# Peer review of "Vitamin D and Stress Fractures in Sport: Preventive and Therapeutic Measures—A Narrative Review"

_medicina, 2021, doi:10.3390/medicina57030223_

Round 1
Reviewer 1 Report
The paper reports a manuscript aiming to reviewing the Vitamin D and stress fractures in sport. Although the manuscript is presented as a systematic review (i.e. the text-flow, searching process, PRISMA diagram, etc.) the paper is presenting a narrative review. Nevertheless, the aim of the manuscript was not defined in the Introduction section neither in the Abstract. The text is well-written and clear to read, the list of references is quite comprehensive and updated and the paper is within the scope of interest of the typical readers of the journal.
However there are several points that deserve some attention from the authors:
Major Comment:
My main concern/major comment is the following: What is the novelty of the present manuscript?
I performed a quick search in PubMed and, among others (see below), I found a manuscript entitled: ‘Vitamin D for Improved Bone Health and Prevention of Stress Fractures: A Review of the Literature’ (PMID: 32516190) published in 2020 (June).
‘Comparing the aims’ between studies (although in the present manuscript the aim was not stated – see comment below) one may observe that both manuscripts (i.e. the aforementioned and the current manuscript) are similar and that there is not a ‘real’ novelty in the present manuscript (at least in the current version). I’m my opinion the manuscript could be improved, thus making it ‘different’ from previous studies, performing a systematic review (and, if possible, meta-analysis).
In this sense, and reinforcing the aforementioned opinion, there are diverse studies with similar aims, e.g. ‘Vitamin d and physical performance’ (PMID: 23657931) published in 2013 in Sports Medicine; PMID: 31382666; PMID: 24256495; PMID: 20970764; etc.
Discretionary comments:
Introduction Section:
The Introduction section is not complete and some aspects were not included/considered.
For example, who are the main contributors to the topic? What did they do and what did the find in their previous studies? And even more important, what novel thing was achieved or did the authors revealed with their manuscript (i.e. which ‘gap’ did they fill in the literature)?
I’m my opinion the Introduction section needs to be re-write to overcome the abovementioned issues as well as to better present the revision of the literature describing and highlighting the weak points (if any) of previous studies and revisions and the special characteristics of interest of the present manuscript… The novelty of their study, what their study add to what is already known… etc.
Finally, what is the objective of the manuscript? The aim was not detailed/described.
Material and Methods:
- Why the authors did not perform a systematic review? And related to this, why the authors did not carry-out a meta-analysis? If there is any ‘objective reason’ this should be stated in the manuscript.
- Although the manuscript is not a systematic review the text is presented as it was. Furthermore, different issues should be clarified: Why was included the PRISMA flow diagram?
- The ‘inclusion criteria’ (as are presented in the current version) are not clear enough and should be better explained to fully understand the rationale of the manuscript for future readers (this is even more important if the text-flow is presented as a systematic review – as is presented in the current version). Further, there is no mention about the subjects’ level and/or training experience that they had, etc. As this fact (e.g. professional vs. recreational sport practitioners) may be important for readers, please inform.
- Why the Web of Science or the Google Scholar databases were not included for the searching process? Please, inform.
- Lines 39-40: I do not totally understand the searching time-frame as is defined in the current version. Do the authors performed the searching from December 2020 to the databases inception? Or do they limits the search only to studies published in 2020? (Looking at the reference list I believe that was from their inception to Dec 2020, but this issue may be defined clearer).
Results and Discussion section:
As the Introduction section was not clear enough and as the main aim of the present manuscript was not clearly defined, is not easy to understand the rationale of the different parts included in this section.
The title of the manuscript is ‘Vitamin D and stress fractures in sport – a narrative review’, however the Results and Discussion are also focused in others aspects (e.g. localization of fractures and/or sex and stress fractures) that were not previously presented as ‘important issues’.
On the other hand, as I mentioned at the beginning, the points addressed in the present manuscript are quite similar to those presented by Lawley et al. 2020 (‘Vitamin D for Improved Bone Health and Prevention of Stress Fractures: A Review of the Literature’ [PMID: 32516190]).
Lastly, in this section the authors may include some notes about how the diverse literature included in the Discussion obtained/measured the Vitamin D, conditions, etc. as this fact may influence the results obtained/derived by the intervention, etc. and thus, the Conclusions derived from the present review-manuscript.
Miscellaneous:
- Please remove the page header “Int. J. Environ. Res. Public Health 2019, 16, x FOR PEER REVIEW” as the Journal is a different one, i.e. Medicina (MDPI).
Author Response
Reviewer 1
The paper reports a manuscript aiming to reviewing the Vitamin D and stress fractures in sport. Although the manuscript is presented as a systematic review (i.e. the text-flow, searching process, PRISMA diagram, etc.) the paper is presenting a narrative review. Nevertheless, the aim of the manuscript was not defined in the Introduction section neither in the Abstract. The text is well-written and clear to read, the list of references is quite comprehensive and updated and the paper is within the scope of interest of the typical readers of the journal.
Answer: We agree with the expert reviewer. Upon submission, the associated editor asked to present the search of literature as a systematic review. Therefore, we inserted the PRISMA diagram which is now deleted. As you have highlighted, we have revised the abstract, introduction and methodology to reflect that the manuscript is a narrative review. We followed the recommendation outlined in Grant and Booth (2009) A typology of reviews: An analysis of 14 review types and associated methodologies. Therefore, we had decided on this methodology as “this method seeks to identify what has been accomplished previously, allowing for consolidation, for building on previous work, for summation, for avoiding duplication and for identifying omissions or gaps.” (pp. 97). We do not wish to misrepresent the methodology and have rectified it. We changed the Abstract to ‘There are numerous risk factors for stress fractures that have been identified in literature. A prolonged lack of vitamin D can lead to stress fractures in athletes since vitamin D insufficiency is associated with an increased incidence of a fracture. A 25-OH vitamin D value of <75.8 nmol/l is a risk factor for a stress fracture. Vitamin D deficiency is, however, only one of many potential risk factors. Well-documented risk factors include female sex, white ethnicity, older age, taller stature, lower aerobic fitness, prior physical inactivity, greater amounts of current physical training, thinner bones, vitamin D deficiency, iron deficiency, menstrual disturbances, and inadequate intake of vitamin D and/or calcium. Stress fractures are not uncommon in athletes and affect around 20% of all athletes. Most athletes with a stress fracture are under 25 years of age. Stress fractures can affect every sporty person, from weekend athletes to top athletes. Stress fractures are common certain sports disciplines such as basketball, baseball, athletics, rowing, soccer, aerobics, and classic ballet. The lower extremity is increasingly affected for stress fractures with the locations of the tibia, metatarsalia and pelvis. Regarding prevention and therapy, vitamin D seems to play an important role. Athletes should have an evaluation of vitamin D-dependent calcium homeostasis based on laboratory tests of 25-OH-D3, calcium, creatinine, and parathyroid hormone. In case of an insufficiency of vitamin D, normal blood levels of ≥ 30 ng/ml may be restored by optimizing the athlete's lifestyle and, if appropriate, oral substitution of cholecalciferol. Very recent studies suggested that the prevalence of stress fractures decreased when athletes are supplemented daily with 800 IU vitamin D and 2000 mg calcium. Recommendations of daily vitamin D intake may go up to 2,000 IU of vitamin D3 per day’.
We added in the Introduction the following ‘Most individuals who experience a stress fracture are young and healthy and do not appear to have an underlying metabolic bone disease [11]. Although no evidence exists that athletes have a higher daily requirement than the general population, vitamin D deficiency in athletes has been linked to a decreased physical performance and a predisposition to stress fractures [12]. Individuals whose bone mineral density is reduced along with a low intake of dietary calcium and low circulating levels of 25-hydroxy vitamin D seem to be at an increased risk of a stress fracture [11]. We have a large knowledge about epidemiology of bone overuse injuries and stress fractures in athletes [13,14], bone health of athletes [15–18], skeletal muscle function in athletes [19] and the influence of vitamin D supplementation on bone mass [16], muscle strength [16] and performance in athletes [17,19–21]. Little is known, however, regarding the relationship between stress fractures in athletes and the importance of vitamin D [6]. Limited experience suggests that calcium and vitamin D supplementation might be helpful [11]. Vitamin D may protect against overuse injuries such as stress fractures [15,20]. The aim of this narrative review is to summarize existing findings about vitamin D in prevention and therapy regarding stress fractures in athletes.’
We have added to the methodology section: “Accordingly, a narrative review was conducted according to the definition outlined by Grant and Booth [22]. This method allowed for identification, summation and consolidation of the existing literature, enabling the discovery of potential gaps or omissions in previous bodies of work.”
However, there are several points that deserve some attention from the authors:
Major Comment:
My main concern/major comment is the following: What is the novelty of the present manuscript?
I performed a quick search in PubMed and, among others (see below), I found a manuscript entitled: ‘Vitamin D for Improved Bone Health and Prevention of Stress Fractures: A Review of the Literature’ (PMID: 32516190) published in 2020 (June).
Answer:
Thank you for your feedback. The objective of this manuscript was specifically examine the role of Vitamin D and stress factors from an athletic/sporting perspective. The reference above discusses a more holistic role of Vitamin D in Bone health, without discussing the implications for sport and athletes in particular. Our review synthesizes the available evidence and provides a sport centric point of view with particular emphasis on the connection between vitamin D and stress fractures in sport especially for the aspects of prevention and therapy.
‘Comparing the aims’ between studies (although in the present manuscript the aim was not stated – see comment below) one may observe that both manuscripts (i.e. the aforementioned and the current manuscript) are similar and that there is not a ‘real’ novelty in the present manuscript (at least in the current version). I’m my opinion the manuscript could be improved, thus making it ‘different’ from previous studies, performing a systematic review (and, if possible, meta-analysis).
Answer: We agree with the expert reviewer and added the following aspect: In this narrative review, we summarize the current state of knowledge about the connection between vitamin D and stress fractures in sport especially for the aspects of prevention and therapy.
We believe that the perspective undertaken in the current review is sufficiently novel for publication as it focuses on sport and athlete health. Although the current manuscript is not a systematic review, we still applied a Systematic search and review approach (as described by Grant and Booth, 2009). Whilst stopping short of a systematic review, for the sake of time and limited funding, a Systematic search and review approach allowed us to generate a “best evidence synthesis” including the identification of gaps and omissions in the literature. Therefore the current manuscript is intended to serve as an up-to-date summation of the available literature that can be used to support future, more comprehensive systematic reviews and meta-analyses.
In this sense, and reinforcing the aforementioned opinion, there are diverse studies with similar aims, e.g. ‘Vitamin d and physical performance’ (PMID: 23657931) published in 2013 in Sports Medicine; PMID: 31382666; PMID: 24256495; PMID: 20970764; etc.
Answer: We agree with the expert reviewer. There are several studies with similar aims. With this in mind, our review drew on these previous studies to present a current and up to date summary of the available evidence. As described in our response above, we intended to create a best evidence synthesis from the available literature. However, in order to more comprehensively focus on sport and athletic implications, we have modified our search have included the following:
. We added now: For the aspects of prevention, we searched the ‘PUBMED’ database with the new terms ‘Prevention stress fractures vitamin D’ resulting in 90 articles. For the aspect of therapy, again using the PUBMED database we searched with the terms ‘therapy of stress fractures vitamin D’ leading to 137 articles.. The newly identified literature was incorporated into the existing manuscript and as a result, providing a more holistic overview of the connection between vitamin D in sport and athletic populations.
Discretionary comments:
Introduction Section:
The Introduction section is not complete and some aspects were not included/considered.
For example, who are the main contributors to the topic? What did they do and what did the find in their previous studies? And even more important, what novel thing was achieved or did the authors revealed with their manuscript (i.e. which ‘gap’ did they fill in the literature)?
Answer: We agree with the expert reviewer and explain in the method section to set the focus on prevention and therapy. Further, we have added the following background literature to the introduction “Athletes who experience a stress fracture are young and healthy and do not have an underlying metabolic bone disease [11]. Although no evidence exists that athletes have a higher daily requirement of vitamin D than the general population, vitamin D deficiency in athletes has been linked to a decreased physical performance and a predisposition to stress fractures [12]. Athletes whose bone mineral density is reduced together with a low intake of dietary calcium and low circulating levels of 25-hydroxy vitamin D seem to be at an increased risk of stress fractures [11]. We have a considerable knowledge about the epidemiology of bone overuse injuries and stress fractures in athletes [13,14], bone health of athletes [15–18], skeletal muscle function in athletes [19] and the influence of vitamin D supplementation on bone mass [16], muscle strength [16] and performance in athletes [17,19–21]. However, little is known regarding the relationship between stress fractures in athletes and the importance of vitamin D [6]. Limited experience suggests that calcium and vitamin D supplementation might be helpful [11]. Vitamin D may protect against overuse injuries such as stress fractures [15,20]. The aim of this narrative review is to summarize existing findings about vitamin D in prevention and therapy regarding stress fractures in athletes.
I’m my opinion the Introduction section needs to be re-write to overcome the abovementioned issues as well as to better present the revision of the literature describing and highlighting the weak points (if any) of previous studies and revisions and the special characteristics of interest of the present manuscript… The novelty of their study, what their study add to what is already known… etc.
Answer: We agree with the expert reviewer and as outlined above, revised the introduction to reflect the context in which the current review was undertake, with special emphasis on what the gaps in the literature are. We have added that our manuscript, attempts to formulate an up-to-date best evidence synthesis of the link between vitamin D and stress fractures in athletes.
Finally, what is the objective of the manuscript? The aim was not detailed/described.
Answer: We agree with the expert reviewer and added in the Introduction the following to explicitly state the objective of the manuscript Although several studies have examined the link between vitamin D and stress fractures in an athletic population, the aim of this narrative review is to summarize existing findings about vitamin D in prevention and therapy regarding stress fractures in athletes and to formulate an up-to-date best evidence synthesis.
We have added a section to the methodology to describe our objective in more detail as well and it read as follows: Several reviews already reported the importance of vitamin D in athletes regarding different aspects such as the influence on performance [20,21], bone health [6], bone health and athletic performance [17], muscle function and performance [19], and the intake of calcium and vitamin D in the prevention of stress fractures [15]. Accordingly, a narrative review was conducted according to the definition outlined by Grant and Booth [22]. This method allowed for identification, summation and consolidation of the existing literature, enabling the discovery of potential gaps or omissions in previous bodies of work. In this narrative review, we summarize the current state of knowledge about the connection between vitamin D and stress fractures in sport especially for the aspects of prevention and therapy. More specifically, we applied a systematic search and review to the literatutre search [22] allowing for a best evidence synthesis.
Material and Methods:
- Why the authors did not perform a systematic review? And related to this, why the authors did not carry-out a meta-analysis? If there is any ‘objective reason’ this should be stated in the manuscript.
Answer: Thank you for your question. We had considered conducting a systematic review and meta-analyses but due to time and funding restrictions, we decided that a narrative review using a systematic search and review approach would be the most effective. Although our review is not the gold standard for conducting reviews, it does draw heavily from the systematic review methodology. Further, the goal of a systematic review is summarize and critically appraise the available literature, however our objective was to create a best evidence synthesis, incorporating multiple study types instead of only focusing on a single study design. From this perspective, we felt the manuscript better reflects the available evidence, drawing from RCT, case control and case study designs. We have included this as part of our limitations.
- Although the manuscript is not a systematic review the text is presented as it was. Furthermore, different issues should be clarified: Why was included the PRISMA flow diagram?
Answer: Thank you for your observation and we agree with you. Upon submission as narrative review, the editor asked to perform a systematic review. However, due to the constraints mentioned previously, we decided to perform a narrative review using the systematic search and review approach. There are similarities between our review methodology and a systematic review, but ultimately, we decided to continue with the current methodology. We have removed the PRISMA flow diagram and have indicated explicitly in the text that the current manuscript is not a systematic review to avoid any confusion.
- The ‘inclusion criteria’ (as are presented in the current version) are not clear enough and should be better explained to fully understand the rationale of the manuscript for future readers (this is even more important if the text-flow is presented as a systematic review – as is presented in the current version). Further, there is no mention about the subjects’ level and/or training experience that they had, etc. As this fact (e.g. professional vs. recreational sport practitioners) may be important for readers, please inform.
Answer: We agree with the expert reviewer and added the following in the methods section:
|
Table 1. Summary of inclusion and exclusion criteria
|
|
|
|
Inclusion criteria |
|
1 |
Studies investigating sport, exercise or physical training |
|
2 |
Investigating the role of vitamin D or make reference to the supplementation, ingestion or use of vitamin D |
|
3 |
Studies investigating bone health, stress fractures, with reference to prevention and/or rehabilitation |
|
4 |
Study designs including RCTs, case control, cross sectional and, case series and reports |
|
5 |
Human participants |
|
|
|
|
|
Exclusion criteria |
|
1 |
Studies in non-athletic or non-physically active populations |
|
2 |
Animal studies |
|
3 |
Study designs including letters, editorials, conference proceedings, short surveys and author responses |
|
|
|
In terms of the level and/or training experience of participants from included studies, where possible we state the level and type of physical activity undertaken. We agree that is an important point and have made made the changes to highlight this.
- Why the Web of Science or the Google Scholar databases were not included for the searching process? Please, inform.
Answer: We agree with the expert reviewer and checked other data bases We have included the following section:
“Ensuring that no relevant publication was missed, we ran web engine searches using Google Scholar and Web of Science databases to screen the first 10 result pages. Any studies that were not part of our initial database search were included.»
- Lines 39-40: I do not totally understand the searching time-frame as is defined in the current version. Do the authors performed the searching from December 2020 to the databases inception? Or do they limits the search only to studies published in 2020? (Looking at the reference list I believe that was from their inception to Dec 2020, but this issue may be defined clearer).
Answer: We agree with the expert reviewer and changed to For this purpose, we searched the 'Scopus' and 'PUBMED' databases for the terms 'stress fracture - vitamin D - athlete' from database inception until January 2021.
Results and Discussion section:
As the Introduction section was not clear enough and as the main aim of the present manuscript was not clearly defined, is not easy to understand the rationale of the different parts included in this section.
Answer: Thank you for your feedback, we have added more background literature to the introduction with the goal of making the manuscript aim more clearly defined.
Further, as we have used a synthesized results and discussion approach of presenting our findings, we have included the following introductory paragraph: A total of 180 articles were included in this review. Thirteen themes were identified and synthesized in the sections that follow.
The title of the manuscript is ‘Vitamin D and stress fractures in sport – a narrative review’, however the Results and Discussion are also focused in others aspects (e.g. localization of fractures and/or sex and stress fractures) that were not previously presented as ‘important issues’.
Answer: We agree with the expert reviewer. To explain better, we add now in the Introduction ‘Athletes who experience a stress fracture are young and healthy and do not have an underlying metabolic bone disease [11]. Although no evidence exists that athletes have a higher daily requirement of vitamin D than the general population, vitamin D deficiency in athletes has been linked to a decreased physical performance and a predisposition to stress fractures [12]. Athletes whose bone mineral density is reduced together with a low intake of dietary calcium and low circulating levels of 25-hydroxy vitamin D seem to be at an increased risk of stress fractures [11]. We have a large knowledge about epidemiology of bone overuse injuries and stress fractures in athletes [13,14], bone health of athletes [15–18], skeletal muscle function in athletes [19] and the influence of vitamin D supplementation on bone mass [16], muscle strength [16] and performance in athletes [17,19–21]. Little is known, however, regarding the relationship between stress fractures in athletes and the importance of vitamin D [6]. Limited experience suggests that calcium and vitamin D supplementation might be helpful [11]. Vitamin D may protect against overuse injuries such as stress fractures [15,20]. The aim of this narrative review is to summarize existing findings about vitamin D in prevention and therapy regarding stress fractures in athletes’ and in the method section ‘Several reviews already reported the importance of vitamin D in athletes regarding different aspects such as the influence on performance [20,21], bone health [6], bone health and athletic performance [17], muscle function and performance [19], and the intake of calcium and vitamin D in the prevention of stress fractures [15]. In this narrative review, we summarize the current state of knowledge about the connection between vitamin D and stress fractures in sport especially for the aspects of prevention and therapy’ and ‘For the specific aspects of prevention, we searched ‘PUBMED’ with the terms ‘prevention - stress – fracture - vitamin D’ leading to 90 articles. For the aspect of therapy, we searched with the terms ‘therapy - stress – fracture - vitamin D’ leading to 137 articles in ‘PUBMED’.
On the other hand, as I mentioned at the beginning, the points addressed in the present manuscript are quite similar to those presented by Lawley et al. 2020 (‘Vitamin D for Improved Bone Health and Prevention of Stress Fractures: A Review of the Literature’ [PMID: 32516190]).
Answer: We thank the expert reviewer for providing this reference. Although the points we present are similar to those of Lawley et al., our review has taken a more structured approach to the reviewing the available literature. Lawley et al. have done a very broad review of the literature (including no description of their methodology) with a very specific focus on the underlying physiology. As stated previously, our approach was to examine the link between vitamin D and stress fractures in an athletic population, the aim of this narrative review is to summarize existing findings about vitamin D in prevention and therapy regarding stress fractures in athletes and to formulate an up-to-date best evidence synthesis. This includes a holistic analysis of the literature in order to summarize and present up-to-date research with a clear methodology that may be included in future systematic reviews and meta-analyses.
Lastly, in this section the authors may include some notes about how the diverse literature included in the Discussion obtained/measured the Vitamin D, conditions, etc. as this fact may influence the results obtained/derived by the intervention, etc. and thus, the Conclusions derived from the present review-manuscript.
Answer: We agree with the expert reviewer and changed the Conclusions to ‘Stress fractures are not uncommon in athletes and affect around 20% of all athletes. Most athletes with a stress fracture are under 25 years of age. Stress fractures can affect every active person, from weekend athletes to top athletes. Stress fractures are common in specific sports disciplines such as basketball, baseball, track and field, rowing, soccer, aerobics, and ballet. The lower extremity is more affected than the upper extremity by stress fractures in the tibia, metatarsals and pelvis. Athletes should have an evaluation of vitamin D-dependent calcium homeostasis based on laboratory tests of 25-OH-D3, calcium, creatinine, and parathyroid hormone. In case of vitamin D insufficiency, normal blood levels of ≥ 30 ng/ml may be restored by optimizing the athlete's lifestyle and, if appropriate, oral substitution of cholecalciferol. Very recent studies suggested that the prevalence of stress fractures decreased when athletes are supplemented daily with 800 IU vitamin D and 2000 mg calcium. Recommendations of daily vitamin D intake may go up to 2,000 IU of vitamin D3 per day’.
Miscellaneous:
- Please remove the page header “Int. J. Environ. Res. Public Health 2019, 16, x FOR PEER REVIEW” as the Journal is a different one, i.e. Medicina (MDPI).
Answer: We agree with the expert reviewer and changed as requested.
Reviewer 2 Report
Thank you for the opportunity to review this manuscript. The narrative review wants to describe the role of vitamin D on stress fractures in athletes. Even if the topic is of interests, different reviews of literature were previously wrote on vitamin D, stress fractures, and sport, the following are examples:
-Tenforde AS, Sayres LC, Sainani KL, Fredericson M. Evaluating the relationship of calcium and vitamin D in the prevention of stress fracture injuries in the young athlete: a review of the literature. PM R. 2010 Oct;2(10):945-9. doi: 10.1016/j.pmrj.2010.05.006. PMID: 20970764.
-Lawley R, Syrop IP, Fredericson M. Vitamin D for Improved Bone Health and Prevention of Stress Fractures: A Review of the Literature. Curr Sports Med Rep. 2020 Jun;19(6):202-208. doi: 10.1249/JSR.0000000000000718. PMID: 32516190.
-Sikora-Klak J, Narvy SJ, Yang J, Makhni E, Kharrazi FD, Mehran N. The Effect of Abnormal Vitamin D Levels in Athletes. Perm J. 2018;22:17-216. doi: 10.7812/TPP/17-216. PMID: 30005732; PMCID: PMC6045510.
-Neal S, Sykes J, Rigby M, Hess B. A review and clinical summary of vitamin D in regard to bone health and athletic performance. Phys Sportsmed. 2015 May;43(2):161-8. doi: 10.1080/00913847.2015.1020248. Epub 2015 Mar 22. PMID: 25797288.
-Watkins CM, Lively MW. A review of vitamin D and its effects on athletes. Phys Sportsmed. 2012 Sep;40(3):26-31. doi: 10.3810/psm.2012.09.1977. PMID: 23528618.
The introduction, according to my opinion, should have to: ii) deeply describe the role of vitamin D, with accurate references; ii) highlight the problems of stress fractures in the sports contest; iii) consider the previous works performed on the topic, bringing the reader to the objective / main goal of the present work.
Related to the methods, I strongly suggest to perform the search also on Web of Science and to use the Boolean Operator AND. Three terms, according to my opinion, are not enough to detect most of the studies that exist in the literature. Examples of other words that could be adopted are stress fractures, stress injuries, athletes, sport, sportsman, sportsmen… Please, adopt PICOS (population, intervention, comparison, outcome, study design) eligibility criteria.
In the results section, according to my opinion, a table / figure could help the reader to better understand the findings. Furthermore, the manuscript lack of discussion. Different sections were dedicated to the results, but no discussion has been properly provided. According to my opinion, the investigators should focalize the manuscript on the relationship between vitamin D and stress fractures (the paragraphs from line 147 and so on are very well done and interesting to read, in line with the title of the paper).
Author Response
Reviewer 2
Thank you for the opportunity to review this manuscript. The narrative review wants to describe the role of vitamin D on stress fractures in athletes. Even if the topic is of interests, different reviews of literature were previously wrote on vitamin D, stress fractures, and sport, the following are examples:
-Tenforde AS, Sayres LC, Sainani KL, Fredericson M. Evaluating the relationship of calcium and vitamin D in the prevention of stress fracture injuries in the young athlete: a review of the literature. PM R. 2010 Oct;2(10):945-9. doi: 10.1016/j.pmrj.2010.05.006. PMID: 20970764.
-Lawley R, Syrop IP, Fredericson M. Vitamin D for Improved Bone Health and Prevention of Stress Fractures: A Review of the Literature. Curr Sports Med Rep. 2020 Jun;19(6):202-208. doi: 10.1249/JSR.0000000000000718. PMID: 32516190.
-Sikora-Klak J, Narvy SJ, Yang J, Makhni E, Kharrazi FD, Mehran N. The Effect of Abnormal Vitamin D Levels in Athletes. Perm J. 2018;22:17-216. doi: 10.7812/TPP/17-216. PMID: 30005732; PMCID: PMC6045510.
-Neal S, Sykes J, Rigby M, Hess B. A review and clinical summary of vitamin D in regard to bone health and athletic performance. Phys Sportsmed. 2015 May;43(2):161-8. doi: 10.1080/00913847.2015.1020248. Epub 2015 Mar 22. PMID: 25797288.
-Watkins CM, Lively MW. A review of vitamin D and its effects on athletes. Phys Sportsmed. 2012 Sep;40(3):26-31. doi: 10.3810/psm.2012.09.1977. PMID: 23528618.
Answer: We agree with the expert reviewer and added all these references. We also looked for more references regarding the aspect of vitamin D considering prevention and therapy of stress fractures and have included the following in the methodology: For the aspects of prevention, we searched the ‘PUBMED’ database with the new terms ‘Prevention stress fractures vitamin D’ leading resulting into 90 articles. For the aspect of therapy, again using the PUBMED database we searched with the terms ‘therapy of stress fractures vitamin D’ leading to 137 articles.
The introduction, according to my opinion, should have to: ii) deeply describe the role of vitamin D, with accurate references; ii) highlight the problems of stress fractures in the sports contest; iii) consider the previous works performed on the topic, bringing the reader to the objective / main goal of the present work.
Answer: We agree with the expert reviewer and added in the Introduction the following text ‘Athletes who experience a stress fracture are young and healthy and do not have an underlying metabolic bone disease [11]. Although no evidence exists that athletes have a higher daily requirement of vitamin D than the general population, vitamin D deficiency in athletes has been linked to a decreased physical performance and a predisposition to stress fractures [12]. Athletes whose bone mineral density is reduced together with a low intake of dietary calcium and low circulating levels of 25-hydroxy vitamin D seem to be at an increased risk of stress fractures [11]. We have a large knowledge about epidemiology of bone overuse injuries and stress fractures in athletes [13,14], bone health of athletes [15–18], skeletal muscle function in athletes [19] and the influence of vitamin D supplementation on bone mass [16], muscle strength [16] and performance in athletes [17,19–21]. Little is known, however, regarding the relationship between stress fractures in athletes and the importance of vitamin D [6]. Limited experience suggests that calcium and vitamin D supplementation might be helpful [11]. Vitamin D may protect against overuse injuries such as stress fractures [15,20]. The aim of this narrative review is to summarize existing findings about vitamin D in prevention and therapy regarding stress fractures in athletes’.
Related to the methods, I strongly suggest to perform the search also on Web of Science and to use the Boolean Operator AND. Three terms, according to my opinion, are not enough to detect most of the studies that exist in the literature. Examples of other words that could be adopted are stress fractures, stress injuries, athletes, sport, sportsman, sportsmen… Please, adopt PICOS (population, intervention, comparison, outcome, study design) eligibility criteria.
Answer: We agree with the expert reviewer and have modified our search strategy. However, as we have not conducted a systematic review, we have revised our methodology to state the current manuscript is a narrative review using a systematic search and review approach. We followed the recommendation outlined in Grant and Booth (2009) A typology of reviews: An analysis of 14 review types and associated methodologies. Therefore, we had decided on this methodology as “this method seeks to identify what has been accomplished previously, allowing for consolidation, for building on previous work, for summation, for avoiding duplication and for identifying omissions or gaps.” (pp. 97). We do not wish to misrepresent the methodology and have rectified it. The PICOS eligibility criteria is generally applied to systematic reviews that critically appraise, group and compare included studies. Whilst stopping short of a systematic review, for the sake of time and limited funding, a Systematic search and review approach allowed us to generate a “best evidence synthesis” including the identification of gaps and omissions in the literature. Therefore, the current manuscript is intended to serve as an up-to-date summation and overview of the available literature that can be used to support future, more comprehensive systematic reviews and meta-analyses.
Further, we have included a table outlining our inclusion and exclusion criteria.
|
Table 1. Summary of inclusion and exclusion criteria
|
|
|
|
Inclusion criteria |
|
1 |
Studies investigating sport, exercise or physical training |
|
2 |
Investigating the role of vitamin D or make reference to the supplementation, ingestion or use of vitamin D |
|
3 |
Studies investigating bone health, stress fractures, with reference to prevention and/or rehabilitation |
|
4 |
Study designs including RCTs, case control, cross sectional and, case series and reports |
|
5 |
Human participants |
|
|
|
|
|
Exclusion criteria |
|
1 |
Studies in non-athletic or non-physically active populations |
|
2 |
Animal studies |
|
3 |
Study designs including letters, editorials, conference proceedings, short surveys and author responses |
|
|
|
In the results section, according to my opinion, a table / figure could help the reader to better understand the findings. Furthermore, the manuscript lack of discussion. Different sections were dedicated to the results, but no discussion has been properly provided. According to my opinion, the investigators should focalize the manuscript on the relationship between vitamin D and stress fractures (the paragraphs from line 147 and so on are very well done and interesting to read, in line with the title of the paper).
Answer: We agree with the expert reviewer and inserted a table with the potential risk factors for stress fractures. We also added more results of literature regarding clinical aspects such as symptoms, diagnosis, treatment and prevention. A separate section is now updated for vitamin D supplementation to prevent stress fractures.
Further, we have included a short paragraph
“A total of 180 articles were included in this review. Thirteen themes were identified and synthesized in the sections that follow.”
This method of combining results and discuss has been done in previous systematic reviews: Hill L, Collins M, Posthumus M. Risk factors for shoulder pain and injury in swimmers: A critical systematic review. Phys Sportsmed. 2015 Nov;43(4):412-20. doi: 10.1080/00913847.2015.1077097. Epub 2015 Sep 14. PMID: 26366502.
This method allows for the contextualization of included studies by discussing the results within the context of the previous literature, allowing for a more holistic overview of the themes that were identified.
Round 2
Reviewer 1 Report
As I mentioned in the previous version, the manuscript is well-written and clear to read, the list of references is quite comprehensive and updated and the paper is within the scope of the journal. Moreover, the manuscript has been improved and the authors have addressed (and properly answered) most of my ‘major concerns’ regarding their manuscript (e.g. justifying the novelty of the present narrative review, among others).
Nevertheless, below, I present a list of minor suggestions/recommendations.
General comment: after the inclusion of the new information (i.e. red text). There are some information that is provided twice (sometimes even more) across the different sections. See an example below:
New text. Lines 430-434. In female Navy recruit volunteers, daily intake of 2000 mg calcium and 800 IU vitamin D reduced the risk of stress fractures significantly [81]. Recommendations of daily vitamin D intake may go up to 2,000 IU of vitamin D3 per day [166]. A target for prevention of stress fractures would be a serum 25(OH)D concentration of 40 ng/mL or greater, achievable with 4000 IU/d of vitamin D3 supplementation [178].
Previous version. Lines 447-448. “A trial on female Navy recruits showed a significant decrease in tibial fractures after daily supplementation of 800 IU vitamin D and 2000 mg calcium [170]“.
I’m my opinion the authors should check this issue to avoid repeating the same information across the article. As the manuscript has valuable and “enough” information regarding vitamin D consumption/intake and stress fractures (in athletes), I strongly believe that this point (i.e. to “focus” and “summarize” the most relevant information) may help and improve the (high) quality of the current manuscript' version.
Discretionary comments:
- Line 25. Stress fractures are common “in” certain sports… add preposition.
- Line 49. Consider to add “normally” in the following sentence: Athletes who experience a stress fracture normally are young and healthy… As stress fractures not occur only in young populations.
- As vitamin D abbreviation was provided in Line 46 (25-OHD), I would like to suggest to the authors to “use the same term for the same concept” through the whole manuscript.
- Risk factors or risk situations for a stress fracture I would like to ask, there exists any cut-off points / thresholds regarding athletes’ bone mineral density measured using whole-body dual-energy X-ray absorptiometry (DXA)? If exists may be an interesting issue to address in the current version of the manuscript (maybe in the aforementioned section or in the Prevention of stress fractures Section – i.e. considering this cut-off point as an “indirect” risk-factor marker).
- Line 253 and 256. Remove extra space after “specificity ,” and after “bone scans may be used”.
- Please check Lines 353-355. Is any information missed? I do not understand well that paragraph.
- Line 383. Have the authors considered mentioning the new Table 2 in this section? As they are presenting information related to intrinsic and extrinsic factors, which were aforementioned (Risk factors or risk situations for a stress fracture Section)
- Lines 430-432: “In female Navy recruit volunteers, daily intake of 2000 mg calcium and 800 IU vitamin D reduced the risk of stress fractures significantly [81].” - Could the authors provide “a number” (e.g., percentage)?
- Line 457. Delete “the” from the following sentence: “during the this review”
- Lines 461-462. Check the following sentence (there are some grammatical mistakes): “we did do Google Scholar and Web of Science searches to ensure we did not miss any relevent studies.”
- Conclusion section. I would suggest to summarize/shortening this section as has a lot of information. I’m my opinion, there are certain information that may be not relevant, e.g. Lines 467-468; “Stress fractures are common in specific sports disciplines such as basketball, baseball, track and field, rowing, soccer, aerobics, and ballet.” Please consider to shortening this section, trying to summarize the most important information provided after performing the review.
Author Response
As I mentioned in the previous version, the manuscript is well-written and clear to read, the list of references is quite comprehensive and updated and the paper is within the scope of the journal. Moreover, the manuscript has been improved and the authors have addressed (and properly answered) most of my ‘major concerns’ regarding their manuscript (e.g. justifying the novelty of the present narrative review, among others).
Nevertheless, below, I present a list of minor suggestions/recommendations.
General comment: after the inclusion of the new information (i.e. red text). There is some information that is provided twice (sometimes even more) across the different sections. See an example below:
New text. Lines 430-434. In female Navy recruit volunteers, daily intake of 2000 mg calcium and 800 IU vitamin D reduced the risk of stress fractures significantly [81]. Recommendations of daily vitamin D intake may go up to 2,000 IU of vitamin D3 per day [166]. A target for prevention of stress fractures would be a serum 25(OH)D concentration of 40 ng/mL or greater, achievable with 4000 IU/d of vitamin D3 supplementation [178].
Previous version. Lines 447-448. “A trial on female Navy recruits showed a significant decrease in tibial fractures after daily supplementation of 800 IU vitamin D and 2000 mg calcium [170]“.
Answer: We agree with the expert reviewer and deleted the second mention. This was once the original study and once from a meta-analysis.
I’m my opinion the authors should check this issue to avoid repeating the same information across the article. As the manuscript has valuable and “enough” information regarding vitamin D consumption/intake and stress fractures (in athletes), I strongly believe that this point (i.e. to “focus” and “summarize” the most relevant information) may help and improve the (high) quality of the current manuscript' version.
Answer: We agree with the expert reviewer and have removed sections that are repetitive. We have also summarized as succinctly as possible the most relevant information.
Discretionary comments:
Line 25. Stress fractures are common “in” certain sports… add preposition.
Answer: We agree with the expert reviewer and changed as requested.
Line 49. Consider to add “normally” in the following sentence: Athletes who experience a stress fracture normally are young and healthy… As stress fractures not occur only in young populations.
Answer: We agree with the expert reviewer and changed as requested.
As vitamin D abbreviation was provided in Line 46 (25-OHD), I would like to suggest to the authors to “use the same term for the same concept” through the whole manuscript.
Answer: We agree with the expert reviewer and added in that section ‘The term 25(OH)D will now be used throughout the manuscript for Vitamin D.’ In some instances, however, we could not always replace (e.g., search strategy).
Risk factors or risk situations for a stress fracture I would like to ask, there exists any cut-off points / thresholds regarding athletes’ bone mineral density measured using whole-body dual-energy X-ray absorptiometry (DXA)? If exists may be an interesting issue to address in the current version of the manuscript (maybe in the aforementioned section or in the Prevention of stress fractures Section – i.e. considering this cut-off point as an “indirect” risk-factor marker).
Answer: We agree with the expert reviewer and added ‘Another approach could be the determination of bone mineral density using dual-energy x-ray absorptiometry to assess the risk for a stress fracture [Am J Sports Med. 1996 Nov-Dec;24(6):810-8; J Bone Miner Res. 1996 May;11(5):645-53; Sports Med. 2019 Jul;49(7):1059-1078]. It has been shown for female athletes that lower bone density was a risk factor for stress fractures in track and field athletes [Am J Sports Med. 1996 Nov-Dec;24(6):810-8.]. Furthermore, bone area and cortical thickness at the tibia were identified as altered both in women with menstrual disturbances and in women with stress fractures [Sports Med. 2019 Jul;49(7):1059-1078]’ in the section ‘Prevention of stress fractures. We found, however, no cut-off values.
Line 253 and 256. Remove extra space after “specificity ,” and after “bone scans may be used”.
Answer: We agree with the expert reviewer and changed as suggested, this occurred after updating the references.
Please check Lines 353-355. Is any information missed? I do not understand well that paragraph.
Answer: We agree with the expert reviewer and changed to ‘Most stress fractures readily heal following a period of modified loading (i.e., reduction in training) and a progressive return to running activities. The high recurrence rate of bone stress injury signals show, however, a need to address their underlying causative factors’ in order to explain better.
Line 383. Have the authors considered mentioning the new Table 2 in this section? As they are presenting information related to intrinsic and extrinsic factors, which were aforementioned (Risk factors or risk situations for a stress fracture Section)
Answer: We agree with the expert reviewer and inserted there Table 2 to refer to that table.
Lines 430-432: “In female Navy recruit volunteers, daily intake of 2000 mg calcium and 800 IU vitamin D reduced the risk of stress fractures significantly [81].” - Could the authors provide “a number” (e.g., percentage)?
Answer: We agree with the expert reviewer and added ‘The calcium and 25(OH)D group had a 20% lower incidence of stress fractures than the control group (5.3% versus 6.6%, respectively)’.
Line 457. Delete “the” from the following sentence: “during the this review”
Answer: We agree with the expert reviewer and changed as suggested, this occurred after updating the references.
Lines 461-462. Check the following sentence (there are some grammatical mistakes): “we did do Google Scholar and Web of Science searches to ensure we did not miss any relevent studies.”
Answer: We agree with the expert reviewer and changed to ‘Thirdly, the current review utilized only two databases to perform our literature search, however, we performed Google Scholar and Web of Science searches to ensure we did not miss any relevant studies’.
Conclusion section. I would suggest to summarize/shortening this section as has a lot of information. I’m my opinion, there are certain information that may be not relevant, e.g. Lines 467-468; “Stress fractures are common in specific sports disciplines such as basketball, baseball, track and field, rowing, soccer, aerobics, and ballet.” Please consider to shortening this section, trying to summarize the most important information provided after performing the review.
Answer: We agree with the expert reviewer and changed to ‘Stress fractures affect around 20% of all athletes where most affected athletes are under 25 years of age. Stress fractures can affect every active person, from weekend athletes to top athletes. Stress fractures are common in specific sports disciplines. The lower extremity is more affected than the upper extremity by stress fractures in the tibia, metatarsals and pelvis. Athletes should have an evaluation of 25(OH)D -dependent calcium homeostasis based on laboratory tests of 25(OH)D, calcium, creatinine, and parathyroid hormone. In the case of 25(OH)D insufficiency, normal 25(OH)D levels of ≥30 ng/ml may be restored by optimizing the athlete's lifestyle and, if appropriate, the oral substitution of 25(OH)D. Very recent studies suggested that the prevalence of stress fractures decreased when athletes are supplemented daily with 800 IU 25(OH)D and 2000 mg calcium. Recommendations of daily 25(OH)D intake may go up to 2,000 IU of 25(OH)D per day. Future studies might investigate the relationship between 25(OH)D and bone mineral density using dual-energy x-ray absorptiometry for athletes with stress fractures and/or athletes with an increased risk of a stress fracture’.
Reviewer 2 Report
Dear Authors,
thank you for the work that you did, your manuscript, according to my opinion, improved importantly. All my questions and comments present appropriate answers and interesting modification directly in the text.
All my major issues have been solved.
There are some minor revision that are still required.
Line 3: “stress fracture” please change in “stress fractures”.
Line 4: “are common certain sports” please change in “are common in certain sports”
Line 20: instead of “female sex” please use “female gender”.
Line 23: please use another term instead of athletes the second time.
Line 25: “Stress fractures are common certain sports disciplines” please add “are common in certain sports”
Line 46: “it has been theorized that vitamin D (25-OHD) has a central role in the development of stress fractures.” Sentence not contextualized with the previous sentences. I suggest to start a new paragraph here.
Line 47-49: “It has been shown that vitamin D deficiency has been associated with an increased incidence of bone fatigue and stress fractures [8–10]” Sentence not contextualized, try to rephrase this part.
Line 71: allowed for indentification,” please change in allowed for identification,”.
Line 76: “allowing for a best evidence” please change in “allowing for the best evidence”.
Line 78: “database inception until the January 2021” please change in “until January 2021”.
Line 133: “physical activity outdoors lead to” please change in “physical activity outdoor leads to”.
Line 199: “ seem to be the larger risk factors of…” please change in “seems to be the larger risk factor for…”
Line 231: “fracture isessential” please change in “fracture is essential”.
Line 254: “they lack specificityn” please change in “they lack in specificity”.
Line 456: “limitations of present review” please change in “limitations of the present review”.
Author Response
Dear Authors,
thank you for the work that you did, your manuscript, according to my opinion, improved importantly. All my questions and comments present appropriate answers and interesting modification directly in the text.
All my major issues have been solved.
There are some minor revision that are still required.
Line 3: “stress fracture” please change in “stress fractures”.
Answer: We agree with the expert reviewer and changed as requested.
Line 4: “are common certain sports” please change in “are common in certain sports”
Answer: We agree with the expert reviewer and changed as requested.
Line 20: instead of “female sex” please use “female gender”.
Answer: Thank you for your comment, however in this context, the appropriate terminology is sex. According to Bartz et al. 2020, sex is a spectrum of biological and physiologic traits characterizing maleness and femaleness. Whereas gender is a continuum of socially constructed roles and behaviors associated with men, women, and gender-spectrum diversity. Therefore, we strongly believe in using the term as we are referring to physiological differences between males and females with regards to their bone metabolism regards of what their specific gender identity is. Bartz, D., Chitnis, T., Kaiser, U. B., Rich-Edwards, J. W., Rexrode, K. M., Pennell, P. B., … Manson, J. E. (2020). Clinical Advances in Sex- and Gender-Informed Medicine to Improve the Health of All. JAMA Internal Medicine. doi:10.1001/jamainternmed.2019.7194
Line 23: please use another term instead of athletes the second time.
Answer: We agree with the expert reviewer and changed to competitors.
Line 25: “Stress fractures are common certain sports disciplines” please add “are common in certain sports”
Answer: We agree with the expert reviewer and changed as requested.
Line 46: “it has been theorized that vitamin D (25-OHD) has a central role in the development of stress fractures.” Sentence not contextualized with the previous sentences. I suggest to start a new paragraph here.
Answer: We agree with the expert reviewer and changed as requested.
Line 47-49: “It has been shown that vitamin D deficiency has been associated with an increased incidence of bone fatigue and stress fractures [8–10]” Sentence not contextualized, try to rephrase this part.
Answer: We agree with the expert reviewer and changed to ‘25(OH)D deficiency has been shown to be associated with an increased incidence of bone fatigue and stress fractures’.
Line 71: allowed for indentification,” please change in allowed for identification,”.
Answer: We agree with the expert reviewer and changed as requested.
Line 76: “allowing for a best evidence” please change in “allowing for the best evidence”.
Answer: We agree with the expert reviewer and changed as requested.
Line 78: “database inception until the January 2021” please change in “until January 2021”.
Answer: We agree with the expert reviewer and changed as requested.
Line 133: “physical activity outdoors lead to” please change in “physical activity outdoor leads to”.
Answer: We agree with the expert reviewer and changed as requested.
Line 199: “ seem to be the larger risk factors of…” please change in “seems to be the larger risk factor for…”
Answer: We agree with the expert reviewer and changed as requested.
Line 231: “fracture isessential” please change in “fracture is essential”.
Answer: We agree with the expert reviewer and changed as requested. This occurred after updating the references.
Line 254: “they lack specificityn” please change in “they lack in specificity”.
Answer: We agree with the expert reviewer and changed as requested. This occurred after updating the references
Line 456: “limitations of present review” please change in “limitations of the present review”.
Answer: We agree with the expert reviewer and changed as requested.